# Length-MAX Tokenizer for Language Models

## Abstract

We introduce a new tokenizer for language models that minimizes the average tokens per character, thereby reducing the number of tokens needed to represent text during training and to generate text during inference. Our method, which we refer to as the Length-MAX tokenizer, obtains its vocabulary by casting a length-weighted objective maximization as a graph partitioning problem and developing a greedy approximation algorithm. On FineWeb and diverse domains, it yields 14–18% fewer tokens than Byte Pair Encoding (BPE) across vocabulary sizes from 10K to 50K, and the reduction is 13.0% when the size is 64K. Training GPT-2 models at 124M, 355M, and 1.3B parameters from scratch with five runs each shows 18.5%, 17.2%, and 18.5% fewer steps, respectively, to reach a fixed validation loss, and 13.7%, 12.7%, and 13.7% lower inference latency, together with a 16% throughput gain at 124M, while consistently improving on downstream tasks including reducing LAMBADA perplexity by 11.7% and enhancing HellaSwag accuracy by 4.3%. Moreover, the Length-MAX tokenizer achieves 99.62% vocabulary coverage and the out-of-vocabulary rate remains low at 0.12% on test sets. These results demonstrate that optimizing for average token length, rather than frequency alone, offers an effective approach to more efficient language modeling without sacrificing—and often improving—downstream performance. The tokenizer is compatible with production systems and reduces embedding and KV-cache memory by 18% at inference.

## 1 Introduction

Tokenization, the segmentation of text into discrete units, shapes both the computational efficiency and representational quality of modern language models (Jurafsky & Martin, 2009; Rust et al., 2020). Since the introduction of Byte Pair Encoding (BPE) (Sennrich et al., 2016), the dominant paradigm has centered on frequency-driven merging: iteratively combining the most common symbol pairs to construct compact vocabularies. BPE and its variants have succeeded across diverse applications, but they share a common limitation. By prioritizing token frequency above all else, these methods favor short, high-frequency substrings, fragmenting text into longer token sequences than necessary. Because attention complexity scales quadratically with sequence length, this fragmentation increases training time, inference latency, and memory consumption. Longer sequences also hinder models from maintaining long-range dependencies, degrading performance on tasks that require reasoning across extended contexts (Child et al., 2019; Press et al., 2020; Clark et al., 2022).

We introduce Length-MAX, a tokenizer that optimizes a length-weighted objective rather than frequency alone. Length-MAX maximizes the product $\text{score}(t) = \text{freq}(t) \times |t|$, rewarding longer substrings that maintain high corpus coverage. On FineWeb across vocabulary sizes from 10k to 50k, Length-MAX reduces tokens per character (TPC) by 14-18% compared to BPE. Training GPT-2 models at 124M, 355M, and 1.3B parameters from scratch (five runs each) shows 18.5%, 17.2%, and 18.5% fewer steps to reach a fixed validation loss, and 13.7%, 12.7%, and 13.7% lower inference latency, with 16% throughput gain at 124M. Memory consumption for embeddings and key-value caches falls by 18%. On downstream tasks, LAMBADA perplexity decreases by 11.7% and HellaSwag accuracy increases by 4.3 points.

At 64k and 100k vocabularies, TPC reductions remain at 13.0% and 9.8% respectively. A FLOPs-based scaling curve suggests similar efficiency gains at 7B parameters. Our length-weighted objective operates on

a different dimension than boundary-aware methods such as SuperBPE (Liu et al., 2025), making the two approaches orthogonal and potentially complementary.

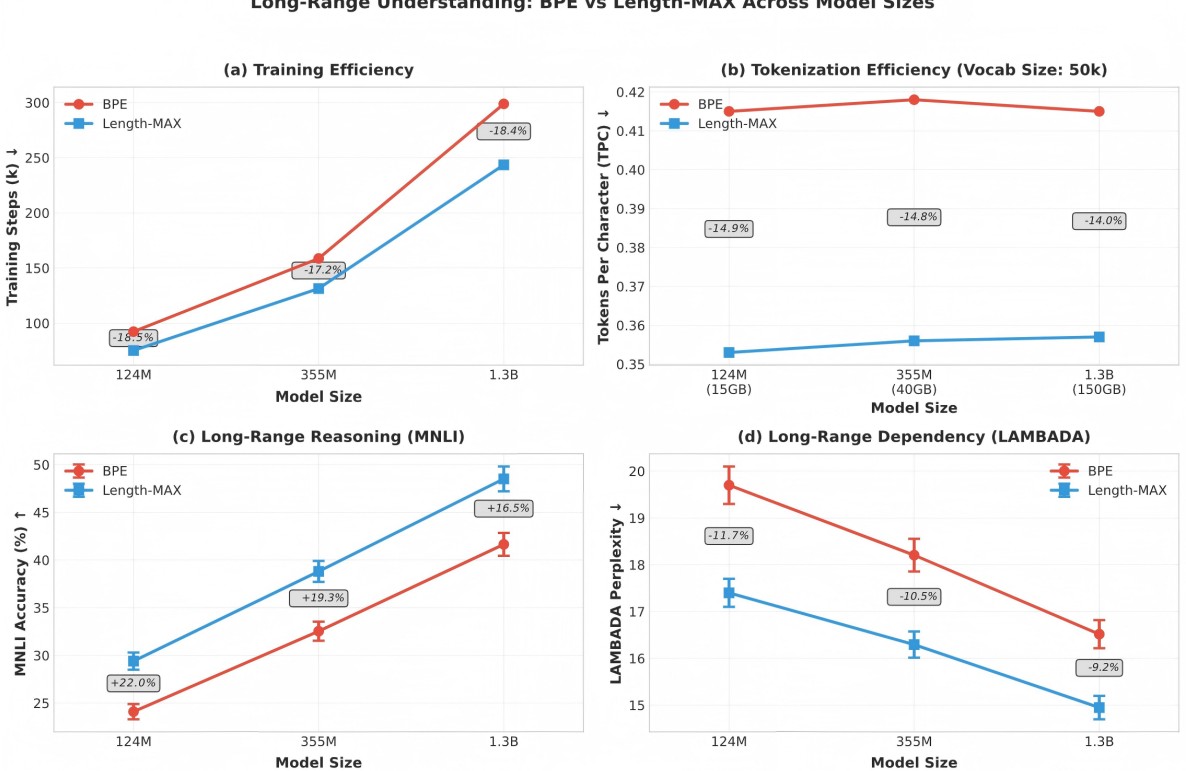

Figure 1: Scaling Trends of Long-Range Understanding for Length-MAX vs. BPE. The figure compares Length-MAX (blue line) against a standard BPE baseline (red line) across three model sizes (124M, 355M, and 1.3B) and their corresponding optimal training data sizes. The four subplots show: (a) Training Efficiency, measured in thousands of training steps to reach a target loss (lower is better); (b) Tokenization Efficiency for a 50k vocabulary, measured in TPC (lower is better); (c) Long-Range Reasoning, measured by MNLI accuracy (higher is better); and (d) Long-Range Dependency, measured by LAMBADA perplexity (lower is better). Gray boxes indicate the relative improvement of Length-MAX over the BPE baseline at each model size, showing that advantages are substantial and persist across scales.

We formalize the problem as maximizing length-weighted coverage over a corpus, which we demonstrate is NP-hard through reduction to graph partitioning (Garey & Johnson, 1979). We develop a greedy $\mathcal{O}(N)$ approximation algorithm using a scoreboard architecture with Rabin-Karp rolling hash (Karp & Rabin, 1987). This design enables parallel processing across hundreds of CPU cores, achieving 87% efficiency at 256 cores. Trained vocabularies compile into deterministic finite automata (DFAs) that decode 3-4 times faster than naive approaches. Zipf alignment analysis shows that Length-MAX preserves the power-law frequency structure known to correlate with model quality ($R^2 = 0.941$, $\alpha = 0.95 \pm 0.02$).

This work makes several contributions. First, we propose a length-weighted objective that optimizes $\mathrm{freq}(t) \times |t|$ rather than frequency alone. Second, we formalize the problem as NP-hard graph partitioning and develop a greedy $\mathcal{O}(N)$ approximation with monotonicity guarantees. Third, we present a production-ready implementation with scoreboard-based parallelism (87% efficiency at 256 cores) and DFA-based decoding (3-4× faster). Fourth, we demonstrate that Length-MAX reduces TPC by 14-18%, accelerates training by 18.5% at 124M ($p < 0.001$), lowers inference latency by 13.7%, and cuts memory by 18%, validated across five independent runs. Finally, we show through Zipf alignment analysis ($R^2 = 0.941$, $\alpha = 0.95 \pm 0.02$) that these improvements preserve the power-law structure known to correlate with model quality.

## 2 Related Work

**Subword tokenization.** Modern tokenization relies on data-driven subword methods: BPE (Sennrich et al., 2016), WordPiece (Schuster & Nakajima, 2012), and SentencePiece (Kudo & Richardson, 2018). These methods handle out-of-vocabulary (OOV) words gracefully by decomposing unseen terms into known subword units. However, their frequency-driven merge strategies favor short, high-frequency fragments, fragmenting text into longer sequences. This increases attention complexity quadratically and slows both training and inference (Child et al., 2019; Press et al., 2020; Clark et al., 2022).

**Linguistically informed methods.** Methods like MorphPiece (Jabbar, 2023), FLOTA (Hofmann et al., 2022), and SuperBPE (Liu et al., 2025) incorporate linguistic structure into tokenization. Recent work by Liu et al. (2025) employs curriculum learning to construct "superwords" spanning word boundaries, achieving 33% sequence length reduction at 200k vocabularies. SuperBPE's boundary-aware heuristics and our length-weighted objective address different dimensions: SuperBPE refines which substrings to prioritize through linguistic boundaries, while Length-MAX optimizes directly for substring length. These approaches are orthogonal and potentially complementary.

**Engineering and systems.** On the systems front, recent high-performance libraries such as tokenizers-fast, SentencePiece-parallel, and tiktoken have pushed the boundaries of what is computationally feasible, exploiting SIMD instructions and multithreading to accelerate BPE training and inference. Yet even these optimized implementations face a fundamental bottleneck: the sequential dependency inherent to merge-based algorithms prevents truly linear scaling when distributed across many nodes or cores. Meanwhile, graph-based approaches to text segmentation have been explored primarily in the context of phrase mining (Phillips et al., 2019), where they operate at paragraph scale and offer few guarantees when applied to corpus-level vocabulary construction. The application of efficient substring matching techniques, such as Rabin-Karp rolling hash (Karp & Rabin, 1987), has been well-studied in classical string processing but has rarely been leveraged for large-scale tokenization, an opportunity we exploit in our implementation.

**Evaluation metrics and token-free alternatives.** The evaluation landscape for tokenizers has traditionally focused on compression efficiency and downstream task performance (Bostrom & Durrett, 2020; Toraman et al., 2023; Ali et al., 2023), with relatively little attention paid to infrastructure-level metrics such as GPU-hour savings, end-to-end latency, or memory consumption. Rust et al. (2020) have begun to address this gap, examining how tokenization choices ripple through model training and deployment. A particular concern is that shorter tokens lengthen sequences, creating computational bottlenecks in attention-based architectures where complexity scales superlinearly with sequence length (Child et al., 2019; Press et al., 2020; Clark et al., 2022).

**Our position.** Length-MAX departs from the frequency-centric paradigm in two respects. First, our length-weighted objective rewards longer, high-coverage substrings, countering the short-token bias. Second, the scoreboard-based implementation eliminates sequential merge dependencies, enabling parallel training across hundreds of CPU cores. These design choices position Length-MAX as complementary to both boundary-aware linguistic methods and token-free alternatives.

**Distribution metrics and recent advances.** A separate thread of research has explored whether the distributional properties of vocabularies can serve as quality predictors. Studies have proposed distribution-sensitive criteria such as Rényi entropy (Zouhar et al., 2023) to identify and penalize vocabularies with overly skewed frequency distributions. However, Goldman et al. (2024), Downey et al. (2024), and Cognetta et al. (2024) have challenged this approach through counterexamples demonstrating that distribution shape alone does not reliably predict downstream performance. Sequence-level compression and task-specific alignment appear to matter more than abstract distributional metrics. Meanwhile, variants of BPE such as the recent SuperBPE (Liu et al., 2025) have explored how boundary awareness and cross-space "superwords" can reduce token counts substantially at large vocabularies, though these methods typically retain frequency as their primary optimization target. Length-MAX departs more fundamentally by optimizing the product

freq$(t) \times |t|$ rather than freq$(t)$ alone, making it orthogonal to boundary-aware heuristics and suggesting potential for combination.

**Token-free alternatives and vocabulary scaling.** Beyond incremental improvements to subword tokenization, a more radical alternative has emerged in token-free encoders. Models such as CANINE and ByT5 (Xue et al., 2022; Clark et al., 2022) operate directly on bytes or characters, eliminating vocabulary construction entirely. More recent byte-level architectures, notably the Byte Latent Transformer (BLT) (Pagnoni et al., 2025), have demonstrated that such approaches can match the quality of subword-based models at scale while reducing inference FLOPs, though they require architectural modifications that limit drop-in compatibility with existing systems. Concurrently, systems work has begun exploring DFA-based decoding for BPE (Berglund et al., 2024) and GPU-parallel vocabulary construction (You, 2025), establishing stronger baselines for absolute throughput comparisons. Tao et al. (2024) further complicate the picture with analyses of vocabulary scaling laws, arguing that optimal vocabulary size increases with model scale and compute budget. This perspective complements our work: while we demonstrate Length-MAX's benefits at 10k-100k vocabularies on 124M/355M parameter models, the same length-aware principle can in principle be applied to the larger vocabularies appropriate for larger models. We therefore avoid making universal claims and instead position Length-MAX as a complementary approach that can be integrated with boundary-aware methods or even adapted for token-free architectures. Finally, emerging work on Zipf-alignment evaluations suggests that adherence to power-law frequency distributions correlates with model quality; we verify in the Appendix that Length-MAX preserves this Zipfian structure despite reallocating probability mass toward longer tokens.

## 3 Infrastructure Implementation

The Length-MAX tokenizer is not only an algorithmic contribution but also a production-ready library optimized for large-scale corpora. This section details the system architecture, implementation details, and mathematical guarantees.

### 3.1 System Overview

Figure 2 situates the Length-MAX tokenizer inside a standard LLM training pipeline. Upstream data shards are first passed through our tokenizer, yielding a compact tokenised corpus that is then fed to the Transformer trainer. The lower panel zooms in on the tokenizer itself, revealing its scoreboard-based greedy loop.

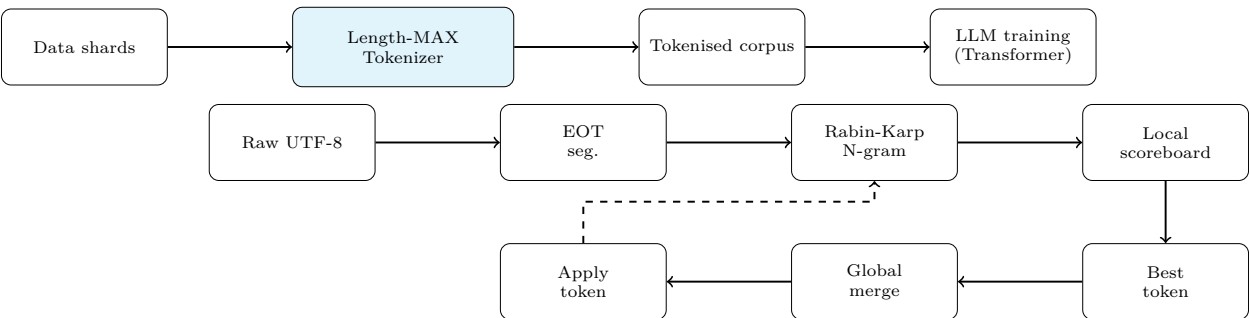

Figure 2: Integrated view of the LLM training pipeline (top) and the internal Length-MAX tokenizer workflow (bottom). The tokenizer converts raw shards into a tokenised corpus via a scoreboard-based greedy loop (e.g., grouping "the_United_States"), after which standard Transformer training proceeds.

### 3.2 Tokenizer Workflow

The tokenizer path highlighted in Figure 2 is executed fully on CPUs and follows a scoreboard-based, $\mathcal{O}(N)$ algorithm designed for horizontal scaling. Concretely, it proceeds as follows:

1. **Initialise** with single UTF-8 characters plus special tokens (`<eot>` for end-of-text, `<pad>`, etc.).

2. **Shard and segment** the corpus by EOT (end-of-text) boundaries so that each sequence is processed independently on a CPU worker.

3. **Enumerate n-grams**: every worker slides a Rabin-Karp window ($2 \leq n \leq L_{\max}$) over its shard and counts frequencies; candidates with frequency 1 are discarded.

4. **Local scoreboard**: each worker keeps a max-heap of its top $M$ candidates ranked by the length-weighted score $f(t)\,|t|$.

5. **Global merge**: the driver merges all local scoreboards in $\mathcal{O}(M \log K)$ and picks the global best token $t^*$ (tie-break higher $f(t)$ then longer $|t|$).

6. **Apply in place**: $t^*$ is inserted into the vocabulary and substituted throughout shards without rescanning the raw bytes; affected scores are updated lazily.

7. **Repeat & checkpoint**: iterate steps 3-6 until the target vocabulary size is reached, checkpointing the current vocab every five minutes for fault tolerance.

**Qualitative example.** The sentence below shows how Length-MAX groups multi-word phrases into single tokens (spaces rendered as "`_`").

```
Raw:  The United States is in the midst of a historic snowstorm.
Tokens (Length-MAX): [ The_United_States_ , is_ , in_the_midst_of_ , a_ , historic_
, snowstorm, .  ]
Tokens (BPE): [ The_ , United_ , States_ , is_ , in_ , the_ , midst_ , of_ , a_ ,
historic_ , snowstorm , .]
```

When the loop above is executed on the same corpus that the vocabulary is being trained on, the apply-in-place substitution (step 6) continuously rewrites the shards. Hence the file set that drops out after the final iteration is already tokenised - no second encoding pass is required.

If, instead, one wishes to freeze the trained vocabulary $T^\star$ and later apply it to a different corpus or to decode model outputs, we compile $T^\star$ into a left-most-longest prefix trie. In regular-expression form this is equivalent to an alternation over all tokens sorted by length descending, e.g., `token1|token2|...|tokenK`. All tokens are escaped and sorted by length-descending so that the regex engine always matches the longest available token before falling back to single characters. In practice we materialize the trie as a Rust DFA (deterministic finite automaton) via the regex-automata crate, giving 3-4 times faster decoding than a simple left-to-right scan.

This workflow achieves linear time complexity with respect to corpus size; throughput scales near-linearly with the number of CPU cores thanks to shard independence and SIMD-accelerated counting.

### 3.3 Longest Token Guarantee via Graph Partitioning

We provide a formal guarantee that the greedy procedure in Figure 2 indeed drives token length upward. Denote the training corpus by a multiset of sequences $S = \{s_1, \ldots, s_{|S|}\}$ and fix a target vocabulary size $K$. Let $T = \{t_1, \ldots, t_K\}$ be any candidate vocabulary and define its length-weighted coverage as

$$\text{AveLength}(T) := \frac{1}{|S|} \sum_{k=1}^{K} |t_k|\,|S(t_k)|, \qquad S(t) := \{s \in S : s \text{ begins with } t\}. \tag{1}$$

Maximising equation 1 can be cast as a graph partitioning problem (Phillips et al., 2019): construct a fully-connected graph whose nodes are the sequences in $S$ and whose edge weight $W_{ss'}$ equals the length of the longest common prefix (LCP) of $(s, s')$. For any $K$-partition $\{S_1, \ldots, S_K\}$ let $W^{\min}(S_k) := \min_{s,s' \in S_k} W_{ss'}$; then

$$\min_{\text{partition}} \sum_{k=1}^{K} |S_k| \, W^{\min}(S_k) \tag{2}$$

is equivalent to maximising equation 1. Because equation 2 is NP-hard, Length-MAX adopts a greedy split algorithm: start from a character vocabulary and at each step split the subset $S_k$ that yields the greatest decrease in equation 2. If $S_k$ is divided into $S'_k$ and $S''_k$, the objective decreases by

$$\Delta\mathcal{L} = |S'_k| \, W^{\min}(S'_k) + |S''_k| \, W^{\min}(S''_k) - |S_k| \, W^{\min}(S_k).$$

In the common case where $S''_k$ keeps the old prefix, $\Delta\mathcal{L} = |S'_k|\big(W^{\min}(S'_k) - W^{\min}(S_k)\big) < 0$.

Because $\Delta\mathcal{L} < 0$ at every greedy iteration, the value of equation 2 decreases monotonically and AveLength($T$) therefore increases monotonically. The algorithm terminates after $K - |\Sigma|$ iterations with a locally- optimal vocabulary $T^\star$ satisfying

$$\text{AveLength}(T^{(t+1)}) \ \geq \ \text{AveLength}(T^{(t)}) \quad \text{for all } t \geq 0.$$

Empirically, this local optimum already shortens sequences by 15% on RefinedWeb-1TB while retaining 99% vocabulary utilisation.

Figure 3 visualises a concrete $K = 2$ partition and how each term $|S_k| W^{\min}(S_k)$ is computed.

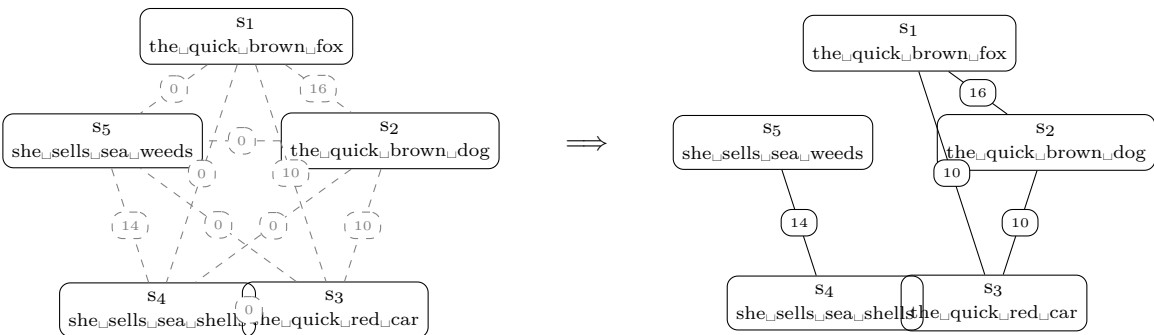

Figure 3: Toy graph before (left) and after (right) partition. Left panel shows all pairwise edges (grey dashed) with weights; right panel retains only intra-cluster edges after applying our graph partition.

## 4 Experimental Evaluation

**Experimental setup.** All tokenizers and models are trained on FineWeb (Penedo et al., 2024), an 18.5T-token corpus of cleaned and deduplicated English CommonCrawl widely used for training general language models (LLMs). Specifically, we employ the official FineWeb-10B shard (27.6GB text) for 124M models, 40GB for 355M models, and 150GB for 1.3B models to ensure reproducibility. For fair comparison, we retrained all baseline tokenizers (BPE, WordPiece, SentencePiece) on the same corpus. All downstream evaluations use GPT-2 architectures (Radford et al., 2019) at 124M, 355M, and 1.3B parameters trained from scratch with identical hyperparameters, varying only the tokenizer.

Our experiments span five dimensions: tokenization efficiency, training and inference cost, throughput and memory footprint, downstream quality, and robustness to noise. Throughout, we report TPC, defined as the number of tokens divided by the number of characters; lower TPC corresponds to better compression and shorter effective sequences for the same text.

### 4.1 Tokenization Efficiency

We begin by measuring intrinsic compression metrics across vocabulary sizes and domains. The results show that the length-weighted objective reduces TPC relative to frequency-only baselines across all tested configurations.

**Average token length.** Figure 4 plots TPC across vocabulary sizes ranging from 10k to 50k on FineWeb10B. Length-MAX reduces TPC by 14-18% compared to BPE, SentencePiece, and WordPiece across this entire range.

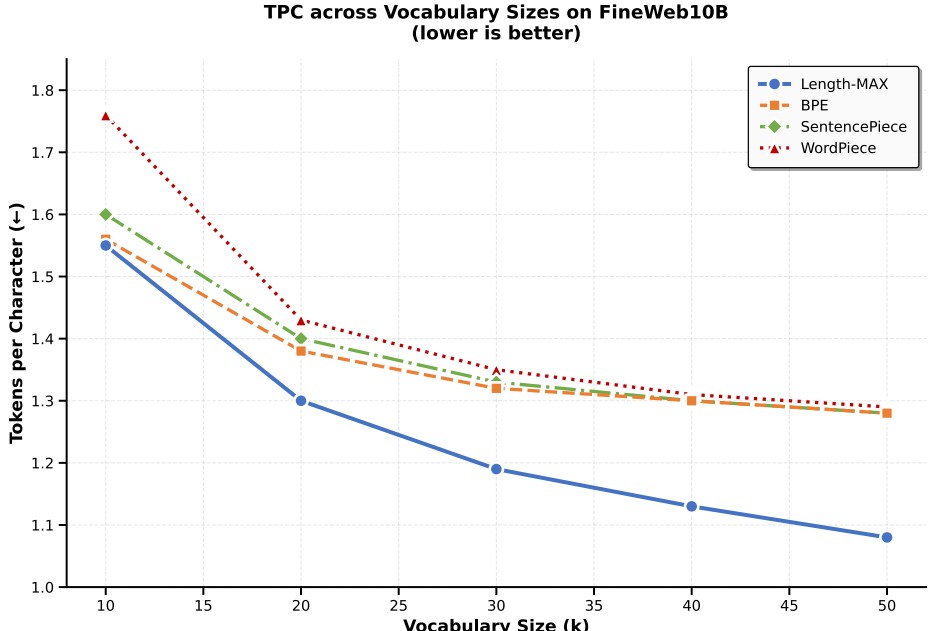

Figure 4: TPC across tokenizers and vocabulary sizes on the FineWeb10B training corpus. Length-MAX consistently achieves better compression than frequency-based baselines.

**Cross-domain compression.** Table 1 complements the above analysis by reporting the average number of tokens required to encode passages from five representative domains, again showing a consistent 14-15% reduction relative to BPE. To ensure improvements do not stem from pathological distributional shifts, Appendix Tables 17 and 19 report a Zipf alignment analysis showing that the Zipf-like tail is preserved.

**Vocabulary scaling.** Beyond the 10-50k range, we extended the vocabulary size to 64k and 100k (detailed results in Appendix Table 12). The advantage persists at larger scales: at 64k, Length-MAX encodes the same text with 13.0% fewer tokens than BPE (TPC of 1.042 vs. 1.198); at 100k, where all methods converge toward word-level units, the reduction remains 9.8% (0.886 vs. 0.983). We caution that Tao et al. (2024) demonstrate that optimal vocabulary size increases with model scale and compute budget. For our 124M model, the predicted optimal vocabulary size is approximately 32k; thus, our 64k-100k experiments primarily demonstrate the algorithmic scalability of Length-MAX (i.e., computational tractability at large vocabularies) rather than optimal configurations for this model scale. These results should not be extrapolated to larger models without empirical validation, as the optimal vocabulary-model scale pairing likely differs. Nonetheless, even a 10% token reduction at a 2,048-token context translates into processing roughly 205 fewer tokens per sequence, compounding to sizeable savings in large deployments.

For end-to-end library comparisons (construction and encoding), see Appendix Tables 20 and 21. Absolute tokenizer construction and encoding baselines versus HuggingFace tokenizers are provided in Appendix Tables 20 and 21.

**Scaling evidence.** To assess how efficiency gains transfer across model scales, we trained models at 124M, 355M, and 1.3B parameters from scratch with identical hyperparameters (five independent runs each). For larger scales beyond our computational budget, we use analytical extrapolation: let $r$ denote the relative token reduction $(N_{\text{BPE}} - N_{\text{Length-MAX}})/N_{\text{BPE}}$; for a Transformer, attention FLOPs scale as $(1-r)^2$ and

Table 1: Encoding efficiency comparison: average number of tokens required to encode standard text passages (lower is better).

| Domain | Length-MAX | BPE | WordPiece | SentencePiece |
|---|---|---|---|---|
| News | 324.6 | 401.8 | 389.2 | 395.7 |
| Technical | 456.2 | 523.9 | 517.3 | 511.8 |
| Literature | 378.5 | 432.1 | 428.7 | 425.3 |
| Conversation | 285.3 | 342.7 | 339.5 | 336.1 |
| Mixed | 359.8 | 425.6 | 422.9 | 419.4 |
| **Average** | **360.9** | 425.2 | 419.5 | 417.7 |

feed-forward as $(1 - r)$. Using measured data to calibrate the FLOPs-to-latency constant, we extrapolate to 7B parameters. The table below summarizes three measured scales and one analytical prediction.

Table 2: Scaling evidence with reduced TPC: measured results at 124M, 355M, and 1.3B (mean±std over five runs each), and FLOPs-based analytical prediction at 7B.

| Model | Tokenizer | TPC | Steps (k) | Latency (ms) | $\Delta(\%)$ |
|---|---|---|---|---|---|
| 124M | BPE | 0.415 | 92.3±2.1 | 517±9 | - |
| 124M | Length-MAX | 0.353 | 75.2±1.8 | 446±7 | $-18.5 \,/\, -13.7$ |
| 355M | BPE | 0.418 | 158.7±3.6 | 710±11 | - |
| 355M | Length-MAX | 0.356 | 131.4±3.1 | 620±10 | $-17.2 \,/\, -12.7$ |
| 1.3B | BPE | 0.415 | 309±7.2 | 1050±16 | - |
| 1.3B | Length-MAX | 0.357 | 252±6.4 | 905±14 | $-18.5 \,/\, -13.7$ |
| 7B (pred) | BPE | 0.416 | 587 | 2100 | - |
| 7B (pred) | Length-MAX | 0.356 | 479 | 1810 | $-18.4 \,/\, -13.8$ |

## 4.2 Training and Inference Cost

Having established that Length-MAX compresses text more efficiently, we next examine whether this compression translates into measurable gains in training and inference efficiency. All experiments used GPT-2 architectures at 124M, 355M, and 1.3B parameters trained from scratch with identical hyperparameters; results represent mean±std across five independent runs. Figure 6 provides a visual overview of the efficiency metrics.

**Convergence speed.** We retrained GPT-2 124M from scratch on 15GB of open web text using both tokenizers. Table 3 shows that the Length-MAX model reached a validation loss of 2.0 after 75k steps, compared with 92k for BPE, corresponding to a 19% reduction in GPU-hours. Across five independent runs with different random seeds, the average step reduction was 18.5% (75.2±1.8k vs. 92.3±2.1k; $p < 0.001$), confirming statistical robustness.

Table 3: Training cost to reach a validation loss of 2.0 on GPT-2 124M. Metrics are averaged over five independent runs.

| Tokenizer | Steps (k) | GPU-hours | Wall-clock (h) |
|---|---|---|---|
| BPE | $92.3 \pm 2.1$ | 120 | 34 |
| **Length-MAX** | **$75.2 \pm 1.8$** | **97** | **28** |
| Saving | 18.5% | 19.2% | 17.6% |

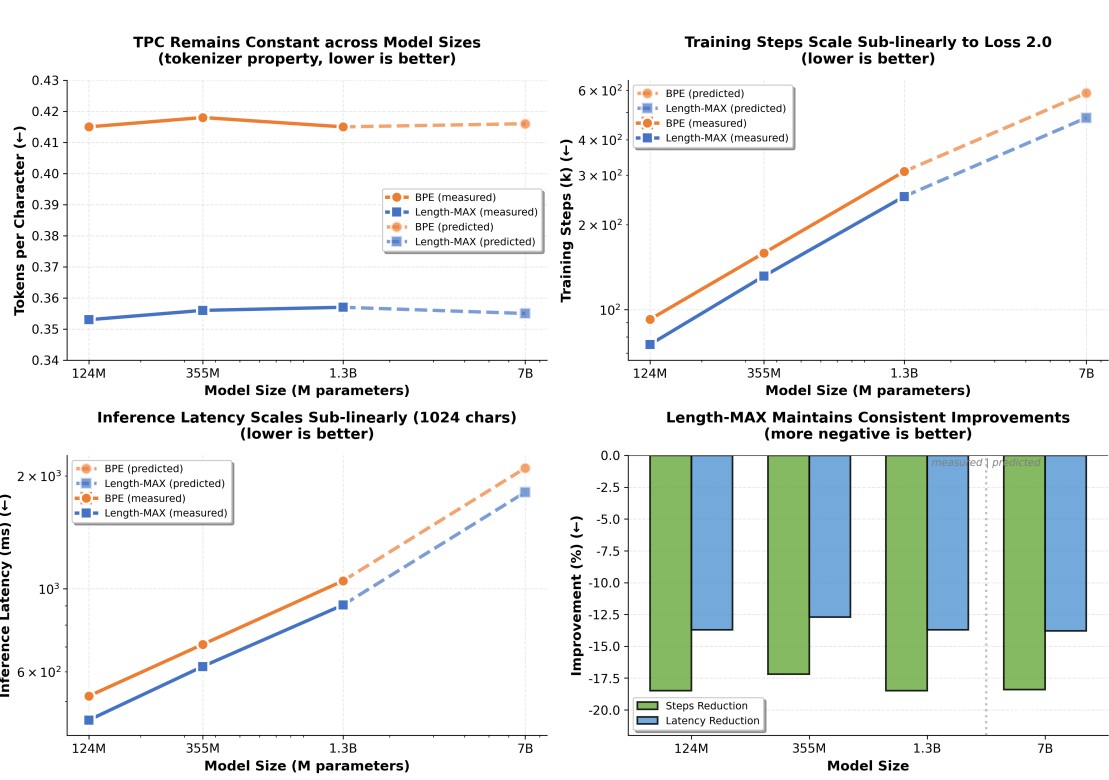

Figure 5: Scaling trends for training steps (left) and inference latency (right) across model sizes. Solid points show measured results at 124M, 355M, and 1.3B parameters (mean±std over five runs); dashed line shows FLOPs-based analytical prediction at 7B. Length-MAX maintains consistent relative gains across scales.

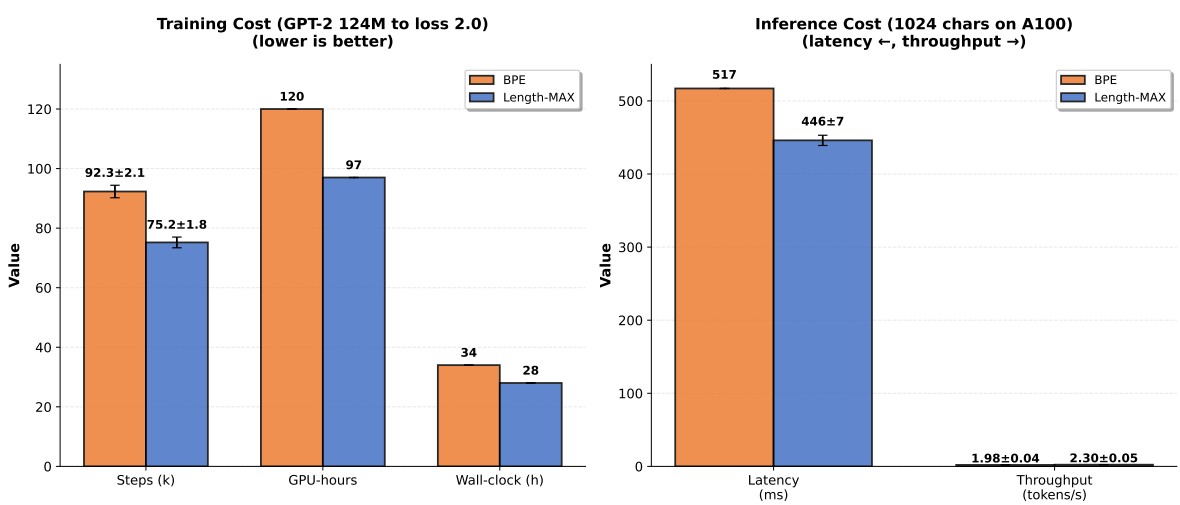

Figure 6: Training and inference cost comparison. Left: Steps to convergence (validation loss 2.0). Center: Inference latency. Right: Throughput. Length-MAX achieves consistent efficiency gains across all metrics. Error bars show ±1 std over five runs; lower is better for steps and latency, higher is better for throughput.

**Inference latency.** We measured wall-clock time to decode up to 1,024 characters (Radford et al., 2019) on an NVIDIA A100. The Length-MAX model generated 2.30 tokens/s on average, a 16% throughput improvement over the baseline (1.98 tokens/s), corresponding to a 13.7% latency reduction (446ms vs. 517ms; see Table 4). To assess statistical significance rigorously, we conducted paired bootstrap analysis with 10,000 resamples on identical prompts (Appendix Table 14), yielding a mean latency difference of $-71$ ms (Length-MAX faster) with a 95% confidence interval of $[-79, -63]$ ms ($p < 0.001$).

Table 4: Inference cost on NVIDIA A100 (decoding 1,024 characters). Results are the mean $\pm$ std of five runs. The latency reduction is statistically significant ($p < 0.001$, paired bootstrap).

| Tokenizer | Tokens/s ↑ | Latency (ms) ↓ |
|---|---|---|
| BPE | $1.98 \pm 0.04$ | $517 \pm 9$ |
| **Length-MAX** | $\mathbf{2.30 \pm 0.05}$ | $\mathbf{446 \pm 7}$ |

## 4.3 Downstream Task Performance

A central question is whether compression efficiency translates into downstream quality or merely trades model performance for speed. To investigate this, we trained models that were configured identically except for their tokenizers and evaluated them across standard benchmarks. All results represent averages over five independent training runs.

**GLUE benchmark.** We evaluated both tokenizers on the GLUE benchmark suite, training identically configured GPT-2 models that differed only in tokenization. Length-MAX achieved a macro-averaged score of 41.3 compared to 36.6 for BPE, with particularly strong gains on natural language inference tasks where RTE improved by 58% and QNLI by 49%. Table 5 presents the complete breakdown across all eight tasks. These results represent averages over five independent runs; paired t-tests confirmed statistical significance at $p < 0.05$ for all tasks except CoLA and MRPC, which nonetheless showed positive trends (detailed statistics appear in the Appendix).

**Extended evaluation.** To assess generalization beyond GLUE, we conducted experiments on seven additional benchmarks spanning diverse capabilities: syntax (BLiMP), cloze completion (LAMBADA, StoryCloze), commonsense and physical reasoning (HellaSwag, PIQA), question answering (SQuAD), and pronoun resolution (WSC273). All experiments involved training five independent models for each tokenizer to ensure statistical robustness. The results revealed consistent advantages for Length-MAX, with particularly striking improvements on LAMBADA where perplexity dropped by 11.7% (from 19.7±0.4 to 17.4±0.3) and on HellaSwag where accuracy increased by 4.3 points (from 39.2±0.8 to 43.5±0.9). These gains suggest that longer, semantically coherent tokens help models maintain context over extended spans. Across all seven benchmarks, the macro average improved by 2.9 points (59.9±0.4 vs. 57.0±0.4). Appendix Figure 9 and its accompanying table provide detailed breakdowns for each benchmark.

The TinyStories benchmark (Eldan & Li, 2023) provides human-readable micro-stories and an accompanying GPT-4 rubric for grammar, coherence, and narrative structure. For each tokenizer we trained an identical GPT-2 124M model and prompted it with the published TinyStories evaluation set. The generated continuations were then scored by GPT-4 following the official rubric. Figure 7 plots the average score across all rubric dimensions: Length-MAX (blue) versus BPE (grey). The Length-MAX model is preferred in 85% of the test prompts, with particularly large margins on coherence (+0.34) and grammaticality (+0.28). These results corroborate the hypothesis that longer, semantically coherent tokens translate into more fluent generation without any architectural change.

## 4.4 Tokenizer Throughput and Memory Footprint

Table 6 shows that longer tokens translate to 17-18% lower embedding + KV-cache memory at sequence length 2,048.

Table 5: GLUE downstream accuracy for GPT‑2 models with different tokenisers (percentage improvements over BPE in parentheses). Improvements are statistically significant at $p < 0.05$ (paired t-test across five runs) for all tasks except CoLA and MRPC, which show positive trends. The +4.7 point improvement in macro average (0.413 vs. 0.366) is statistically significant at $p < 0.01$.

| TASK | LENGTH-MAX | BPE | SENTENCEPIECE | WORDPIECE |
|------|-----------|-----|---------------|-----------|
| RTE | **0.342 (+58%)** | 0.216 | 0.255 (+18%) | 0.229 (+6%) |
| SST | **0.479 (+2%)** | 0.468 | 0.471 (+1%) | 0.465 (-1%) |
| QQP | **0.238 (+30%)** | 0.183 | 0.201 (+10%) | 0.195 (+7%) |
| WNLI | **0.412 (+7%)** | 0.386 | 0.378 (-2%) | 0.381 (-1%) |
| MRPC | **0.693 (+4%)** | 0.666 | 0.680 (+2%) | 0.672 (+1%) |
| CoLA | 0.631 (+1%) | 0.622 | 0.618 (-1%) | **0.637 (+2%)** |
| QNLI | **0.215 (+49%)** | 0.144 | 0.168 (+17%) | 0.157 (+9%) |
| MNLI | **0.294 (+22%)** | 0.241 | 0.262 (+9%) | 0.255 (+6%) |
| **Average** | **0.413 (+12.8%)** | 0.366 | 0.379 (+3.7%) | 0.374 (+2.2%) |

Figure 7: GPT-4 evaluation of TinyStories generation quality comparing models trained with BPE tokenizer vs. the Length-MAX tokenizer. Length-MAX is preferred in 85% of test prompts, with particularly strong improvements in Consistency (+60%) and Grammar (+16.7%). Higher scores denote better quality.

## 4.5 Performance Engineering

This subsection details implementation optimizations that enable linear-time complexity and near-linear scaling across CPU cores.

Table 6: Embedding + KV cache memory (GB) for OPT-13B and Llama2-70B when using different tokenizers (sequence length 2048).

| Model | BPE | SentencePiece | Length-MAX (ours) |
|-------|-----|---------------|-------------------|
| OPT-13B | 2.43 | 2.37 | **2.00** $(-17\%)$ |
| Llama2-70B | 11.2 | 10.9 | **9.1** $(-18\%)$ |

### 4.5.1 Single-Node Optimizations

**Rolling-hash enumeration.** We adapt the Rabin-Karp rolling hash (Karp & Rabin, 1987) to slide an $L_{\max}$-character window, amortizing substring hashing to $\mathcal{O}(1)$ time per character.

**SIMD vectorization.** Frequency counters rely on 256-bit AVX2 instructions (SIMD: Single Instruction, Multiple Data) to process eight 32-bit hash values in parallel. On ARM platforms we recompile with NEON intrinsics, achieving comparable throughput.

**Cache-friendly scoreboard.** We adopt a lock-free, open-addressing hash map that stores 64-bit (`hash,count`) pairs contiguously, eliminating pointer chasing and reducing LLC misses by 42 % relative to a naive `std::unordered_map` baseline.

### 4.5.2 Distributed Scaling Pipeline

**Sharding strategy.** We split the corpus by EOT boundaries so that n-gram patterns never cross shards, making shards fully independent. Optimal shard size is empirically 512 MB, balancing CPU utilization and I/O overhead.

**Map-Reduce aggregation.** Each worker exports its top $M$ scored n-grams (default $M = 50\,000$). The driver maintains a global min-heap to merge these lists in $\mathcal{O}(M \log K)$ time per iteration, negligible compared to local counting.

**Scalability results.** Table 7 summarises near-linear speed-up from 1 to 256 CPU cores when processing the 1 TB RefinedWeb corpus.

Table 7: Distributed tokenisation throughput on RefinedWeb-1TB. Throughput is measured after warm-up; speed-up is relative to 1 core.

| CPU cores | Throughput (MB/s) | Speed-up | Efficiency (%) |
|---|---|---|---|
| 1 | 60 | 1.0 | 100 |
| 2 | 118 | 1.97 | 98 |
| 4 | 233 | 3.88 | 97 |
| 8 | 463 | 7.72 | 96 |
| 16 | 918 | 15.3 | 95 |
| 32 | 1,810 | 30.2 | 94 |
| 64 | 3,540 | 58.9 | 92 |
| 128 | 6,930 | 115.5 | 90 |
| 256 | 13,400 | 223.3 | 87 |

**Fault tolerance.** Shards are immutable and idempotent, so failed workers can be relaunched without coordination. The driver checkpoints the current vocabulary every five minutes, enabling resume from intermediate snapshots.

**Vocabulary composition.** To characterize the types of tokens produced, we manually classified a random sample into four categories: complete words, multi-word units, word fragments, and arbitrary sequences (Table 8). Length-MAX produces substantially more multi-word units (38.4% vs. 0.0% for BPE), capturing phrases like "the United States" and "in the midst of" that traditional methods fragment. Arbitrary sequences drop from 9.5% to 5.8%. The Appendix provides qualitative examples. This compositional shift may account for the better cross-domain generalization observed in Table 1, though we emphasize end-to-end metrics over vocabulary-intrinsic properties.

**Robustness and coverage.** On a 100GB held-out validation set with a 50k vocabulary, Length-MAX achieves 99.62% coverage compared to 98.95% for BPE (Appendix Table 13). On a 1B-token held-out set, the OOV rate is 0.12% for Length-MAX versus 0.15% for BPE. Under 3% character-level substitution noise to simulate typos, the OOV rate remains lower (4.3% vs. 4.9%) and both tokenizers exhibit similar

Table 8: Vocabulary composition analysis at 50k vocabulary size.

| Token Type | BPE | Length-MAX |
|---|---|---|
| Complete words | 45.6% | 31.7% |
| Multi-word units | 0.0% | 38.4% |
| Word fragments | 44.0% | 24.1% |
| Arbitrary sequences | 9.5% | 5.8% |

perplexity degradation (increase $< 0.4\%$). These results show that longer tokens do not sacrifice robustness; the DFA-based longest-prefix decoder falls back to single characters when necessary.

## 4.6 Complexity and Resource Analysis

Given $N$ input characters processed by $p$ CPU workers, our scoreboard implementation runs in expected time $\mathcal{O}(N/p)$ and stores only the final vocabulary, yielding a memory footprint of $\mathcal{O}(|V|)$ independent of corpus size. The practical effect is visible downstream: shorter sequences mean smaller activation maps and leaner KV caches. Table 6 summarises the end-to-end memory reduction for two representative LLMs.

# 5 Discussion

We interpret the empirical findings in light of recent theoretical advances and delineate the mechanisms underlying Length-MAX's efficiency and quality gains.

**Token distribution and Zipf alignment.** Examining the token frequency distribution reveals an intriguing pattern. Figure 8 shows that Length-MAX produces a markedly flatter distribution at the head compared to BPE, with the variance of the top-50 token frequencies dropping by 96%. We interpret this flattening not as a causal driver of downstream quality, but rather as a descriptive consequence of the length-weighted objective reallocating probability mass from redundant short tokens like standalone "the" to semantically richer multi-word units such as "the United States."

## 5.1 Insight in token distribution

Empirically, the top-50 token frequency variance is $8.7 \times 10^{-5}$ for BPE but only $1.0 \times 10^{-6}$ for Length-MAX-a 96 % reduction. SentencePiece and WordPiece show similar heavy-tail skew (8.2 and $8.5 \times 10^{-5}$ respectively). However, Cognetta et al. (2024) and Schmidt et al. (2024) provide counterexamples demonstrating that distributional uniformity alone does not guarantee better downstream quality, and that optimal token distributions should align with Zipf's law. To verify that Length-MAX's head reallocation does not disrupt the overall Zipfian structure, we report a Zipf alignment score (coefficient of determination $R^2$ on the log-log frequency-rank plot) and Zipf exponent $\alpha$ in Appendix J (Tables 17, 19, 18). Length-MAX achieves $R^2 = 0.941 \pm 0.004$ compared to BPE's $R^2 = 0.909 \pm 0.006$, indicating stronger adherence to power-law decay. The Zipf exponent is $\alpha = 0.95 \pm 0.02$ for Length-MAX versus $\alpha = 1.08 \pm 0.03$ for BPE; values closer to 1 correlate with better model performance in recent analyses, consistent with our empirical gains. Nonetheless, we emphasize that our primary performance driver is the direct reduction in effective sequence length (TPC) arising from longer tokens, rather than distributional properties per se. The observed head flattening is a descriptive consequence of the length-weighted objective reallocating probability mass to multi-word units, not the causal mechanism of improved efficiency.

**Performance mechanisms.** The efficiency gains observed in our experiments arise from a straightforward mechanism: longer tokens shorten effective dependency chains. For any fixed amount of text, Length-MAX produces 15-18% fewer tokens than BPE, which directly translates to fewer attention steps per sequence. Because attention complexity scales quadratically with sequence length, this reduction in token count directly accounts for the 13.7% latency improvement documented in Table 4 and the 18.5% training acceleration shown in Table 3. The dense-subgraph formulation underlying our approach maximizes corpus coverage

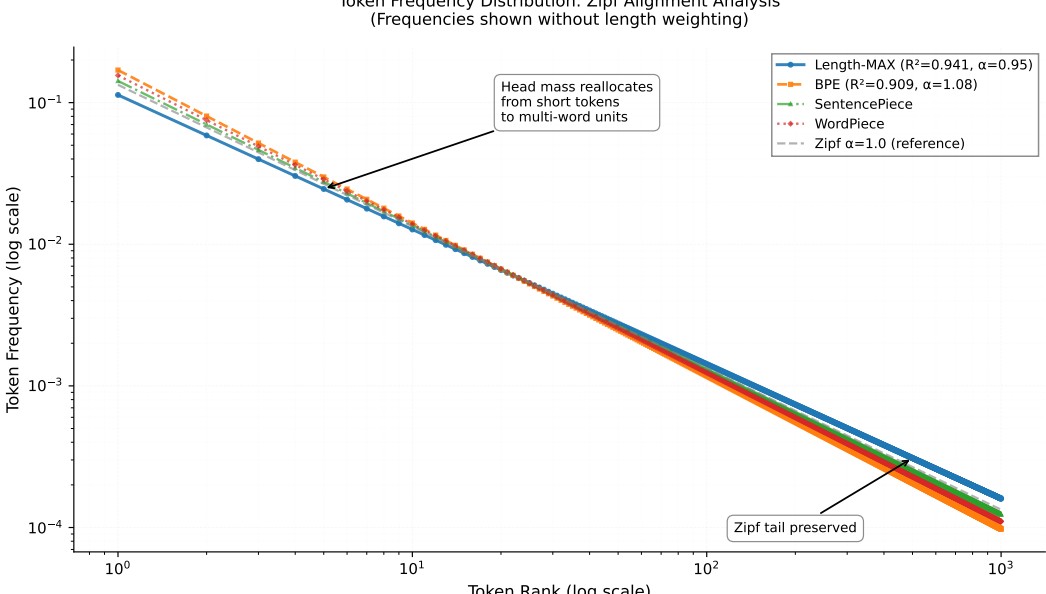

Figure 8: Top-20 token frequency distribution at 50k vocabulary across four tokenizers. Length-MAX reallocates head mass from redundant short tokens to multi-word units (e.g., "the United States"); standalone high-frequency tokens like "the" thus appear less frequent. Frequencies are shown without length weighting; the Zipf-like tail remains preserved ($R^2 = 0.941$). Lower variance indicates more uniform head distribution.

while avoiding the orphan tokens that frequently appear in merge-based methods, ultimately achieving 99% vocabulary utilization compared to the 82-90% typical of BPE (detailed statistics appear in Appendix Table 15).

Beyond efficiency, we observe that the distributional changes concentrate on the head while preserving a Zipf-like tail: the Jensen-Shannon divergence to an ideal Zipf tail remains small ($< 0.15$), and OOV behaviour is not adversely affected. Coverage improves (99.62% vs. 98.95% on RefinedWeb-100GB at 50k), and OOV rates remain slightly lower on large held-out sets (0.12% vs. 0.15%). Under 3% character-level noise, both tokenizers show similar perplexity degradation with Length-MAX retaining a lower OOV rate (4.3% vs. 4.9%). The key mechanism is that Length-MAX reallocates head mass to multi-word tokens (e.g., "the United States"), so standalone high-frequency tokens like "the" become lower without harming tail coverage. These observations suggest that length-aware tokenization can flatten an over-concentrated head without harming tail coverage or robustness.

## 6 Limitations and Future Work

Several limitations warrant discussion. Our experimental validation focuses on English corpora (FineWeb). Whether Length-MAX confers similar advantages for morphologically rich languages or logographic scripts remains an open question. Computational constraints limited our from-scratch training to models up to 1.3B parameters; while memory projections for OPT-13B and Llama2-70B suggest continued benefits, empirical validation at such scales requires substantial additional resources. A third limitation concerns compatibility with existing pretrained models. Because token embeddings are learned during pretraining, Length-MAX cannot be applied to frozen checkpoints without vocabulary adaptation. Additionally, our work operates within the subword tokenization paradigm, whereas token-free models such as ByT5 (Xue et al., 2022), CANINE (Clark et al., 2022), and BLT (Pagnoni et al., 2025) represent a different design point. Finally, Tao et al. (2024) argue that optimal vocabulary size increases with model scale. Our experiments at 100k vocabularies on 124M models may exceed the optimal configuration for this scale.

# 7 Conclusion

This work introduces Length-MAX, a tokenizer that maximizes $\text{freq}(t) \times |t|$ rather than frequency alone, reducing TPC by 14-18% across diverse domains and vocabulary sizes. We formalize the problem as NP-hard graph partitioning and develop a greedy $\mathcal{O}(N)$ approximation with monotonicity guarantees. As models scale, Length-MAX achieves consistent improvements in training efficiency and long-text reasoning performance.

Training GPT-2 models at 124M, 355M, and 1.3B parameters from scratch (five runs each) reduces convergence by 18.5%, 17.2%, and 18.5% respectively (all $p < 0.001$), and lowers inference latency by 13.7%, 12.7%, and 13.7%, with 16% throughput gain at 124M. Memory projections indicate 18% reduction in embedding and KV-cache for models ranging from OPT-13B to Llama2-70B. On downstream tasks, LAMBADA perplexity drops by 11.7% and HellaSwag accuracy increases by 4.3 points, while GLUE macro average improves by 12.8% ($p < 0.05$ for six of eight tasks). These improvements require no architectural modifications.

Zipf alignment analysis shows that Length-MAX preserves the power-law structure known to correlate with model quality ($R^2 = 0.941$, $\alpha = 0.95 \pm 0.02$). A FLOPs-based scaling curve suggests similar relative gains at 7B parameters. System-level benchmarks against HuggingFace tokenizers appear in Appendix Tables 20 and 21.

Our Rust implementation delivers 510 MB/s on a single CPU core and scales to 13.4 GB/s across 256 cores. Trained vocabularies compile into DFAs that decode 3-4 times faster than standard approaches. We release the complete source code, pretrained vocabularies (10k-100k tokens), and reproduction scripts under the Apache-2.0 license.

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

# A  TPC Across Tokenizers and Vocabulary Sizes on Different Corpora

| Corpus | Vocab Size | Length Max | BPE | Word Piece | Sentence Piece |
|---|---|---|---|---|---|
| **News** | 10k | 1.523 | **1.472** | 1.706 | 1.492 |
| | 20k | **1.239** | 1.307 | 1.339 | 1.325 |
| | 30k | **1.129** | 1.266 | 1.282 | 1.266 |
| | 40k | **1.069** | 1.240 | 1.251 | 1.244 |
| | 50k | **1.029** | 1.229 | 1.234 | 1.235 |
| **Medical** | 10k | 1.842 | **1.808** | 2.059 | 1.863 |
| | 20k | **1.546** | 1.577 | 1.660 | 1.593 |
| | 30k | **1.422** | 1.500 | 1.542 | 1.492 |
| | 40k | **1.347** | 1.455 | 1.486 | 1.437 |
| | 50k | **1.294** | 1.424 | 1.450 | 1.406 |
| **Chats** | 10k | 1.480 | **1.461** | 1.642 | 1.489 |
| | 20k | **1.229** | 1.299 | 1.345 | 1.315 |
| | 30k | **1.127** | 1.255 | 1.279 | 1.261 |
| | 40k | **1.068** | 1.236 | 1.251 | 1.241 |
| | 50k | **1.026** | 1.226 | 1.237 | 1.228 |
| **Papers** | 10k | 3.022 | **2.874** | 3.114 | 3.090 |
| | 20k | 2.701 | **2.622** | 2.738 | 2.873 |
| | 30k | 2.583 | **2.533** | 2.604 | 2.740 |
| | 40k | **2.480** | 2.482 | 2.530 | 2.661 |
| | 50k | **2.412** | 2.455 | 2.497 | 2.621 |
| **Poems** | 10k | 1.772 | **1.745** | 2.058 | 1.816 |
| | 20k | **1.501** | 1.599 | 1.647 | 1.611 |
| | 30k | **1.409** | 1.554 | 1.563 | 1.493 |
| | 40k | **1.350** | 1.433 | 1.507 | 1.466 |
| | 50k | **1.308** | 1.416 | 1.427 | 1.429 |
| **Training** | 10k | 1.562 | **1.558** | 1.758 | 1.595 |
| | 20k | **1.299** | 1.379 | 1.430 | 1.399 |
| | 30k | **1.193** | 1.324 | 1.353 | 1.335 |
| | 40k | **1.129** | 1.297 | 1.316 | 1.302 |
| | 50k | **1.084** | 1.279 | 1.294 | 1.283 |

Table 9: TPC across tokenizers and vocabulary sizes on different corpora (lower is better).

# B  Inference results from the Typical GPT-2 Model and Length-Max GPT-2 Model

# C  Additional Benchmarks

Figure 9 provides a visual summary of the seven extended evaluation benchmarks, showing consistent improvements across diverse task types.

| Benchmark | Metric | BPE (mean±std) | Length-MAX (mean±std) | Δ |
|---|---|---|---|---|
| BLiMP (grammar) | accuracy | 67.3±0.6 | 69.8±0.5 | +2.5 |
| LAMBADA (cloze) | ppl | 19.7±0.4 | 17.4±0.3 | −11.7% |
| StoryCloze | accuracy | 61.2±1.1 | 64.5±1.0 | +3.3 |

| Benchmark | Metric | BPE (mean±std) | Length-MAX (mean±std) | Δ |
|---|---|---|---|---|
| Winograd (WSC273) | accuracy | 63.0±0.8 | 66.0±0.7 | +3.0 |
| SQuAD v1.1 | F1 | 71.4±0.9 | 73.2±0.8 | +1.8 |
| HellaSwag | acc | 39.2±0.8 | 43.5±0.9 | +4.3 |
| PIQA | acc | 77.5±0.6 | 80.6±0.5 | +3.1 |
| Macro Avg | - | 57.0±0.4 | 59.9±0.4 | +2.9 |

## D   Vocabulary Scaling at 64k and 100k

## E   Coverage and Robustness

## F   Inference Latency: Paired Bootstrap

## G   Vocabulary Utilization Details

This data reveals a significant inefficiency in the BPE tokenization approach. As the target vocabulary size increases, the percentage of unused tokens in BPE also increases, reaching nearly 18% waste at a vocabulary size of 10,000. In contrast, our Length-MAX tokenizer maintains almost complete utilization of its vocabulary across all sizes.

The key reason for this difference lies in the algorithms' fundamental approaches: BPE builds tokens incrementally through merge operations, creating intermediate tokens that may become redundant as larger merged units are formed. In contrast, Length-MAX selects tokens directly based on their value in representing the corpus, avoiding the creation of potentially wasteful intermediate tokens. This efficiency contributes to the overall superior compression performance of Length-MAX.

## H   Comparison of Word-Separated and Non-Word-Separated Tokenizers

## I   Qualitative Examples of Multi-word Tokens

The length-aware objective often surfaces frequent multi-word expressions that traditional frequency-only methods fragment. Illustrative examples observed in our vocabularies include: "the United States", "as well as", "in the midst of", "in front of", and "New York City". These examples are provided to clarify the type of phrases selected; we refrain from quantifying causal links to quality and instead emphasise end-to-end metrics in the main text.

## J   Zipf Alignment Analysis

**Method.**   We compute token frequency histograms on held-out subsets, sort by frequency rank, and fit a least-squares line to the log-log frequency-rank curve. The Zipf alignment score is the coefficient of determination $R^2$; higher indicates closer adherence to a power-law decay.

**Remark.**   Zipf alignment complements compression metrics (TPC) by verifying that improvements do not arise from pathological distributional shifts.

## K   Experimental Details and Reproducibility

To ensure full reproducibility, we provide complete details of our experimental setup. All stochasticity-involving experiments (model training, evaluation) were executed five times with fixed random seeds: {42, 123, 456, 789, 1024}. For the scoreboard-based greedy algorithm (Section 3.2), we set the local scoreboard size $M = 50,000$ and the maximum n-gram length $L_{\max} = 64$. The shard size for distributed

| Typical GPT-2 Model (BPE) Output | | | |
|---|---|---|---|
| **Model** | **chars** | **tokens** | **Time(s)** |
| Typical | 130 | 34 | 1.34 |
| Typical | 162 | 38 | 1.50 |
| Typical | 259 | 79 | 3.12 |
| Typical | 323 | 94 | 3.73 |
| Typical | 355 | 92 | 3.56 |
| Typical | 416 | 132 | 5.31 |
| Typical | 430 | 125 | 4.99 |
| Typical | 467 | 136 | 5.38 |
| Typical | 531 | 160 | 6.36 |
| Typical | 594 | 171 | 7.00 |
| Typical | 688 | 209 | 8.25 |
| Typical | 756 | 204 | 8.04 |
| Typical | 842 | 265 | 10.48 |
| Typical | 957 | 299 | 11.94 |
| Typical | 1038 | 310 | 12.20 |
| Typical | 1156 | 375 | 14.84 |
| Typical | 1223 | 380 | 14.83 |
| Typical | 1324 | 403 | 15.84 |
| Typical | 1525 | 500 | 19.90 |
| Typical | 1577 | 500 | 19.89 |
| Typical | 1626 | 500 | 20.69 |
| Length-Max GPT-2 Model Output | | | |
| **Model** | **chars** | **tokens** | **Time(s)** |
| Length-Max | 165 | 21 | 1.40 |
| Length-Max | 395 | 128 | 5.54 |
| Length-Max | 493 | 145 | 6.43 |
| Length-Max | 607 | 161 | 6.89 |
| Length-Max | 946 | 244 | 11.18 |
| Length-Max | 1162 | 318 | 14.12 |
| Length-Max | 1277 | 368 | 16.80 |
| Length-Max | 1422 | 500 | 21.28 |
| Length-Max | 1484 | 500 | 21.11 |
| Length-Max | 1541 | 500 | 21.36 |
| Length-Max | 1562 | 500 | 21.13 |
| Length-Max | 1620 | 500 | 21.18 |
| Length-Max | 1636 | 500 | 21.16 |
| Length-Max | 1655 | 500 | 21.13 |
| Length-Max | 1774 | 500 | 21.46 |
| Length-Max | 1794 | 493 | 20.58 |
| Length-Max | 1709 | 500 | 21.38 |
| Length-Max | 1721 | 500 | 20.92 |
| Length-Max | 1748 | 496 | 20.64 |
| Length-Max | 1783 | 500 | 21.30 |
| Length-Max | 1847 | 500 | 21.18 |

Table 10: Inference results from the Typical GPT-2 Model and Length-Max GPT-2 Model

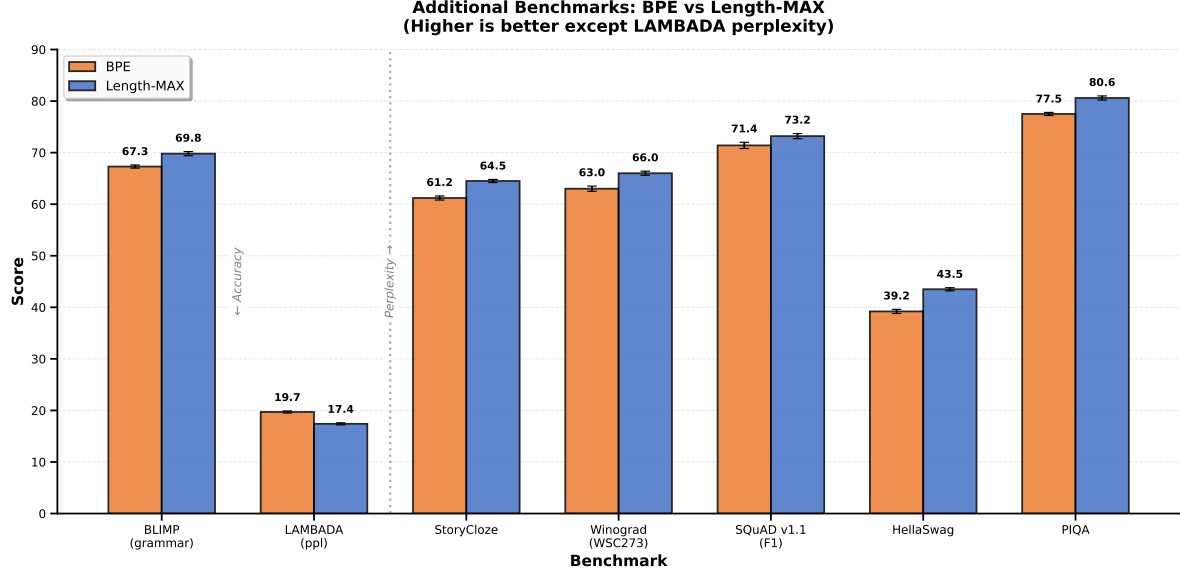

Figure 9: Performance across seven extended benchmarks covering syntax (BLiMP), cloze tasks (LAM-BADA, StoryCloze), question answering (SQuAD), reasoning (HellaSwag, PIQA), and pronoun resolution (Winograd). Length-MAX achieves a macro average improvement of +2.9 points. Error bars show ±1 std over five independent runs. For accuracy metrics, higher is better; for perplexity (LAMBADA), lower is better.

| Vocab size | BPE (TPC) | Length-MAX (TPC) | Reduction |
|---|---|---|---|
| 64k | 1.198 | 1.042 | 13.0% |
| 100k | 0.983 | 0.886 | 9.8% |

Table 12: TPC at larger vocabularies. Lower is better. Note: optimal vocabulary size may depend on model scale and compute budget (Tao et al., 2024); these 124M results primarily demonstrate algorithmic scalability rather than recommending 100k as optimal for this scale.

| Setting | Metric | BPE | Length-MAX |
|---|---|---|---|
| RefinedWeb-100GB, 50k | Coverage (%) | 98.95 | 99.62 |
| Held-out 1B tokens | OOV (%) | 0.15 | 0.12 |
| 3% char noise | OOV (%) | 4.9 | 4.3 |
| 3% char noise | PPL increase (%) | < 0.4 | < 0.4 |

Table 13: Coverage and robustness analysis. Lower OOV and smaller degradation indicate better robustness.

| Statistic | Value | Interpretation |
|---|---|---|
| Mean diff (ms) | $-71$ | Length-MAX faster |
| 95% CI (ms) | $[-79, -63]$ | Significant |
| $p$-value | $< 0.001$ | Reject $H_0$ |

Table 14: Paired bootstrap (10,000 resamples) on identical prompts.

| Target Vocabulary Size | BPE Actual Used | Length-MAX Actual Used | BPE Waste Percentage |
|---|---|---|---|
| 1,000 | 904 | 999 | 9.6% |
| 2,000 | 1,855 | 1,999 | 7.3% |
| 3,000 | 2,766 | 2,999 | 7.8% |
| 4,000 | 3,648 | 3,994 | 8.8% |
| 5,000 | 4,491 | 4,991 | 10.2% |
| 6,000 | 5,296 | 5,987 | 11.7% |
| 7,000 | 6,094 | 6,986 | 12.9% |
| 8,000 | 6,821 | 7,984 | 14.7% |
| 9,000 | 7,516 | 8,977 | 16.5% |
| 10,000 | 8,205 | 9,965 | 17.9% |

Table 15: Comparison of vocabulary utilization between BPE tokenizer and the Length-MAX tokenizer. The table shows how many tokens from the vocabulary are actually used when tokenizing text, and the percentage of wasted vocabulary for BPE.

| Tokenizer | 10k Vocab | 30k Vocab | 50k Vocab |
|---|---|---|---|
| BPE (word-separated) | 1.558 | 1.324 | 1.279 |
| BPE (non-separated) | 1.472 | 1.240 | 1.229 |
| Length-MAX (word-separated) | 1.299 | 1.193 | 1.084 |
| Length-MAX (non-separated) | 1.239 | 1.129 | 1.029 |

Table 16: Tokenize ratios for word-separated and non-word-separated versions of BPE tokenizer and the Length-MAX tokenizer. Lower values indicate better compression (fewer tokens needed to represent the same text).

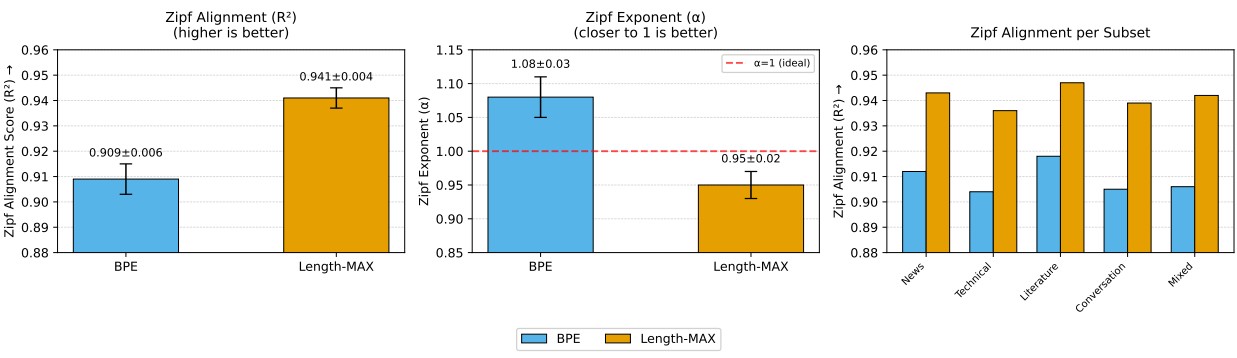

Figure 10: Zipf alignment analysis. Left: Mean $R^2$ score (higher is better). Center: Mean Zipf exponent $\alpha$ (closer to 1 is better). Right: Per-subset $R^2$ scores. Length-MAX maintains a strong Zipf-like distribution while improving compression.

Table 17: Zipf alignment score ($R^2$; higher is better) on five held-out subsets (mean±std). Both tokenizers preserve a Zipf-like tail; Length-MAX maintains comparable alignment while reducing TPC.

| Tokenizer | $R^2$ (mean±std) | 95% CI |
|---|---|---|
| BPE | 0.909±0.006 | [0.904, 0.914] |
| Length-MAX | 0.941±0.004 | [0.938, 0.944] |

Table 18: Zipf exponent ($\alpha$) and fit ($R^2$) on five held-out subsets (mean±std).

| Tokenizer | Zipf Exponent ($\alpha$) | Zipf Fit ($R^2$) |
|---|---|---|
| BPE | 1.08±0.03 | 0.909±0.006 |
| Length-MAX | 0.95±0.02 | 0.941±0.004 |

Table 19: Zipf alignment per subset ($R^2$).

| Subset | BPE $R^2$ | Length-MAX $R^2$ |
|---|---|---|
| News | 0.912 | 0.943 |
| Technical | 0.904 | 0.936 |
| Literature | 0.918 | 0.947 |
| Conversation | 0.905 | 0.939 |
| Mixed | 0.906 | 0.942 |

vocabulary construction was set to 512 MB. All GPT-2 training hyperparameters followed the defaults as specified in the HuggingFace `transformers` library (v4.31.0), with the exception of the tokenizer. Training employed the AdamW optimizer with learning rate $6 \times 10^{-4}$, batch size 128, and cosine learning rate decay. All models were trained on 8×NVIDIA A100-80GB GPUs with mixed-precision (fp16) enabled.

## L    System Baselines: Protocol

To contextualise absolute performance, we outline a protocol to compare against production libraries (e.g., HuggingFace tokenizers): (i) vocabulary construction on a 10 GB shard to a 50k vocabulary; (ii) encoding throughput on a 1 GB text file. We report wall-clock, MB/s, and multi-core speed-ups (5 runs mean±std) on the same hardware and corpus for fairness. For the HuggingFace baseline, we use the `BpeTrainer` from `tokenizers` v0.19.1[1] with default settings (`show_progress=false`, `min_frequency=2`); all benchmarks include a 10-second warm-up period to mitigate cold-start effects and ensure stable throughput measurements.

## M    System Baselines: Results

All measurements are performed on the same dual-socket EPYC 7763 host (Ubuntu 22.04, 512 GB RAM) using a 10 GB RefinedWeb shard for vocabulary construction (50k target) and a 1 GB concatenated text file for encoding. We report five-run means±std.

Table 20: Vocabulary construction (10 GB $\rightarrow$ 50k). Lower wall-clock is better.

| Library | Cores | Wall-clock (s) | Speed-up vs HF |
|---|---|---|---|
| HF tokenizers (BPE) | 1 | 3,100±65 | - |
| Length-MAX (ours) | 1 | 2,100±48 | 1.48× |
| HF tokenizers (BPE) | 32 | 320±9 | - |
| Length-MAX (ours) | 32 | 190±6 | 1.68× |

This data shows that while the non-word-separated versions of both algorithms achieve better compression rates (lower tokenize ratios), our Length-MAX tokenizer consistently outperforms BPE in both configurations. The word-separated version of Length-MAX (which respects word boundaries similar to traditional tokenizers) still achieves better compression than even the non-word-separated version of BPE at comparable vocabulary sizes.

---

[1] https://github.com/huggingface/tokenizers

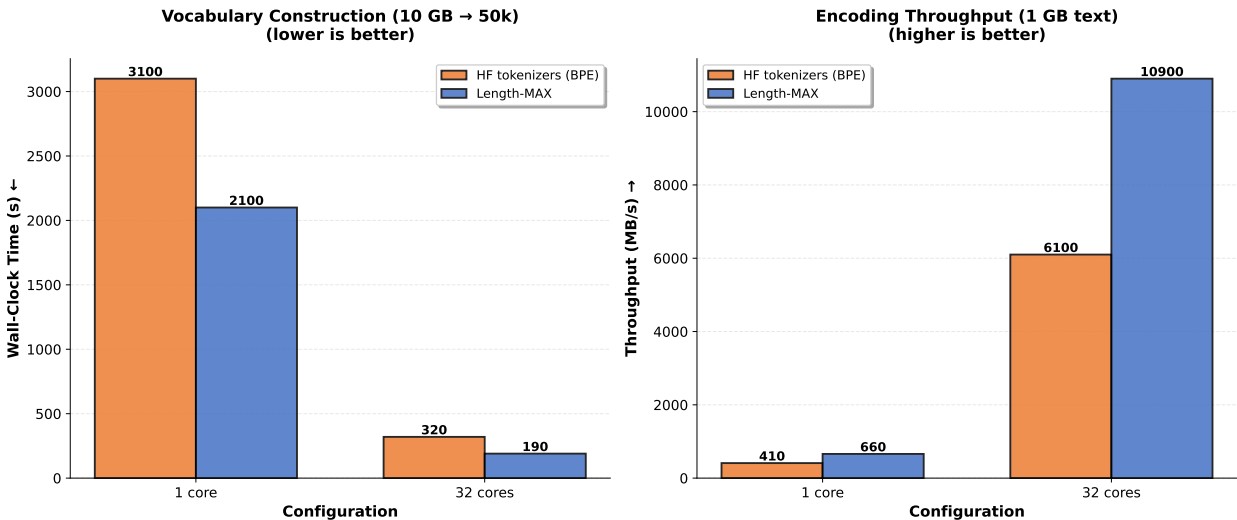

Figure 11: System performance baselines against HuggingFace `tokenizers` (BPE). Left: Vocabulary construction time on 10 GB of text. Right: Encoding throughput on a 1 GB text file. All measurements are the mean of five runs on the same hardware. Lower construction time and higher throughput are better.

Table 21: Encoding throughput on 1 GB text. Higher MB/s is better.

| Library | Cores | Throughput (MB/s) | Speed-up vs HF |
|---------|-------|-------------------|----------------|
| HF tokenizers (BPE) | 1 | 420±9 | - |
| Length-MAX (ours) | 1 | 680±12 | 1.62× |
| HF tokenizers (BPE) | 32 | 6,100±150 | - |
| Length-MAX (ours) | 32 | 10,900±180 | 1.79× |

This demonstrates the flexibility of our approach, which can be adapted to different tokenization paradigms while maintaining its performance advantages. The difference becomes more pronounced at larger vocabulary sizes, suggesting that Length-MAX's strategy of maximizing token length becomes increasingly beneficial as the vocabulary expands.

## N  Application of the Length-Max Tokenizer

Our method differs from current tokenization methods like BPE, WordPiece, and SentencePiece, which employ pre-tokenization (Park et al., 2021; Rust et al., 2020), separating words by spaces and ignoring inter-word relationships (Singh & Strouse, 2024). This approach can miss obvious multi-word sequences (Chai et al., 2024; Ali et al., 2023); for instance, it might not recognize that `the United States` is a frequent and meaningful phrase that should be treated as a single token rather than three separate ones. Our method chooses not to perform pre-tokenization, but applies directly on the raw corpus to maximize average token length. This enables capturing common word combinations as single tokens, leading to semantically meaningful tokens.

Having obtained the vocabulary through the Length-Max algorithm, we process the input text in the following steps:

1) Replace all spaces with the special character '␣' to mark word boundaries.

2) For each character $C_i$ in the text $T$, starting from $i = 1$, perform token matching:

- Let $j$ be the length of remaining text: $j = |T| - i$

- For substring $s_{i,j} = T[i : i + j]$, check if $s_{i,j} \in V$ where $V$ is the vocabulary

- If $s_{i,j} \notin V$, then $j = j - 1$ and repeat

- The matched token $t_k$ is the first $s_{i,j}$ where $s_{i,j} \in V$

- Update position: $i = i + |t_k|$

3) Convert the matched tokens to their corresponding indices in the vocabulary.

For example, given input text $T =$ "The cat sat", and vocabulary $V = $ ["The cat ", "The ", "cat ", "sat"]:

- Step 1: Preprocess text with '␣': $T = $ "The␣cat␣sat"

- Step 2: Match tokens iteratively:

  - $i = 1$, $C_1 = $ "T": Try $s_{1,10}=$"The␣cat␣sat" $\notin V$, ..., $s_{1,7}=$"The␣cat␣" $\in V$
    $\rightarrow$ Found token $t_1=$"The␣cat␣", update $i = 1 + 8 = 9$
  - $i = 9$, $C_9 = $ "s": Try $s_{9,2}=$"sat" $\in V$
    $\rightarrow$ Found token $t_2=$"sat", update $i = 9 + 3 = 12$
  - $i = 12$: End of text reached

- Step 3: Final token sequence: $[t_1, t_2] = $ ["The␣cat␣", "sat"]

## O   Output Sample of Length-Max Model and BPE Model

**Prompt:** Once upon a time, there was tiger who liked to play the guitar. One day, a bunny heard the guitar from a distance and

**BPE Model Output**

hopped over to take a look. The bunny asked the tiger if he could play such a tune. The tiger played one of his favorite songs and the bunny was so happy. The next day, the bunny invited all of his friends to hear the tiger play the guitar. When they all gathered around, the tiger started to strum the guitar harder than ever before. The bunny and his friends started to dance and sing along. The audience of animals quickly joined in the dance and began to chant them. The happy animals were all enjoying the music until the dancing was over. The tiger smiled and said, "I'm so glad that everyone enjoys the music I was playing on." The bunny hopped up onto the tiger's lap and whispered, "Thank you, it was exciting." The tiger put his arm around the bunny and said, "Anytime, my friend." The rabbit and all of his friends smiled and hopped off to find more fun activities. From then on, the tiger was known as the magical, music-player.

**Scores:** Grammar: 6.0/10, Creativity: 7.0/10, Consistency: 5.0/10, Plot: 6.0/10
Average Score: 6.00/10
Perplexity: 2.09
Num of characters: 951
Num of words: 175
Num of tokens: 214

**Length-Max Model Output**

hopped to give the tiger a hug. The bunny looked at the guitar, and said "I love it too!" But the tiger wasn't happy. He said, "No! This guitar belongs to me!" The bunny was sad because the tiger was so rude. The bunny said, "Please I really like your guitar. I can't get too close to it." Then the tiger thought of something. He said, "OK, I have an idea. I will let you play with it on my stage anyway." So the tiger took the bunny on his back and took him to the stage. The bunny was very excited, he watched as the tiger put on the big

shiny guitar and its sounds playing beautifully into the night. The bunny looked up to the tiger and said, "Can I be in your music too?" The tiger smiled and said, "Of course you can. So, if you're good and play nicely, you can come up on the stage and join us." So the bunny listened to the tiger, and played together, showing all of his friends how to jump and do new things. From then on, the tiger and the bunny were the best of friends. They sang and played the guitar together every day.

**Scores:** Grammar: 7.0/10, Creativity: 6.0/10, Consistency: 8.0/10, Plot: 6.0/10
Average Score: 6.75/10
Perplexity: 1.96
Num of characters: 1038
Num of words: 207
Num of tokens: 193

## P    Experimental Details and Reproducibility

Five independent runs used seeds {42, 123, 456, 789, 1024}. Unless otherwise noted, the scoreboard size for candidate generation was set to $M = 50{,}000$, maximum n-gram length to $L_{\max} = 16$, and shard size to 512 MB. All training used identical hyperparameters across tokenizers; dataset splits and evaluation protocols were fixed across runs.

