# OpenReview forum: "Length-MAX Tokenizer for Language Models"
_TMLR — Decision pending for TMLR_

### Review · Reviewer_woJf · 2025-12-08

**Summary Of Contributions:**

The authors propose a new tokenization algorithm based on a length-weighted frequency objective. They describe their method and provide theoretical justification. Then they apply their method for GPT-2 training and compare it with BPE, WordPiece, and SentencePiece. The method showed strong gains against these baselines, improving efficiency and downstream performance.

**Strengths**
1. The motivation is clear. The problem seems relevant for the community.
2. The conceptual justification is clear.
3. The reported gains seem substantial.

**Weaknesses**
1. The theoretical justification does not seem to be directly related to the algorithm of the desired objective.
2. While BPE is a widespread baseline, the paper lacks an empirical comparison with newer tokenization methods.

**Audience:**

Yes

**Audience Explanation:**

Tokenization seems a promising avenue for improving the efficiency of LLM training.

**Broader Impact Concerns:**

I do not see a need for a Broader Impact Statement.

**Claims And Evidence:**

No

**Claims Explanation:**

While the empirical performance of the method is relatively strong, the theoretical portion of the paper is quite confusing. I will present all my concerns below.

1. (minor) While I understand the rationale behind the $f(t) |t|$ heuristic, does not the introduction of a token shorten a text by $f(t) (|t| - 1)$?
2. How are TPC and AveLength metrics related? While I understand the rationale for optimizing TPC, I do not see why we might be interested in AveLength, since it concerns only sentence prefixes in the corpus, not the corpus length after tokenization.
3. Similarly, what is the relation between the procedure in Section 3.2 and the greedy procedure in Section 3.3? I do not see a direct correspondence between optimizing for all n-grams and optimizing for sentence prefixes.
4. (minor) While the empirical evaluation seems strong, it does not involve more efficient tokenizers that have appeared since BPE (e.g., SuperBPE).

**Requested Changes:**

1. Please define $n$, $L_{max}$, and $K$ in Section 3.2.
2. Please clarify the link between Sections 3.2 and 3.3.
2. (minor) Please compare the current method with state-of-the-art tokenizers.

---

### Review · Reviewer_qUkg · 2025-12-14

**Summary Of Contributions:**

This paper introduces Length-MAX, a tokenizer that optimizes freq(t)×|t| instead of frequency alone. The authors formalize the problem as NP-hard graph partitioning and develop a greedy O(N) approximation algorithm with monotonicity guarantees. Experiments on GPT-2 (124M, 355M, 1.3B) demonstrate 14-18% reduction in tokens per character, 18.5% fewer training steps, 13.7% lower inference latency, and improvements on downstream tasks including LAMBADA and HellaSwag.

**Audience:**

Yes

**Audience Explanation:**

Tokenization is fundamental to LLM. The length-weighted objective is simple and has practical potential for reducing training and inference costs and improve performance.

**Broader Impact Concerns:**

No ethical concerns.

**Claims And Evidence:**

No

**Claims Explanation:**

(1)	The paper only compares against BPE, WordPiece, and SentencePiece, but omits SuperBPE (Liu et al., 2025) which is discussed in related work and claims 33% sequence reduction—stronger than Length-MAX's 14-18%. The authors state these methods are "orthogonal and potentially complementary" but provide no empirical comparison or combination experiments.

(2)	The downstream task improvements lack causal analysis—it is unclear whether gains come from shorter sequences, more semantically coherent tokens, or both.

(3)	All experiments use GPT-2. The OPT-13B and Llama2-70B results in Table 6 are only projections. Add more modern models will be more convincing.

(4)	The paper claims efficiency as a big advantage, while no discussion of how Length-MAX interacts with modern efficiency methods like vLLM, Flash Attention, or linear attention.

**Requested Changes:**

Critical:
•	Add comparison with SuperBPE under identical experimental settings
•	Validate on at least one modern architecture (e.g., Llama-2-7B or others)
•	Provide ablation or analysis explaining why downstream performance improves
Suggested:
•	Discuss interaction with modern inference frameworks (vLLM, Flash Attention)

---

### Review · Reviewer_sK8o · 2026-01-06

**Summary Of Contributions:**

The submission introduces Length-MAX, a tokenizer construction method that explicitly optimizes for fewer tokens per character by favoring merges/substrings that are both frequent and long (informally, scoring candidates proportional to frequency times length). The paper motivates a length-weighted “coverage” objective and provides a graph-partitioning view, along with a greedy procedure that monotonically improves the surrogate objective.

Empirically, across multiple vocabulary sizes (10K–64K) and GPT-2 model scales (124M/355M/1.3B), the paper reports substantial reductions in token count relative to BPE (e.g., 14–18% at 10K–50K, ~13% at 64K) and corresponding gains in training/inference efficiency, including fewer steps/GPU-hours to reach a fixed validation loss and faster generation for fixed character budgets.

The work also includes a detailed, production-oriented tokenizer implementation (CPU sharding, rolling hash, SIMD, map-reduce candidate aggregation, checkpointing) with strong throughput scaling to many cores, and it evaluates downstream task performance (GLUE and other benchmarks) reporting consistent improvements over several baselines.

Key strengths: (i) clear efficiency-driven motivation and substantial token reduction, (ii) broad evaluation with multiple seeds and significance tests, (iii) strong systems/implementation evidence for tokenizer construction.

Key weaknesses: (i) the paper does not precisely define the sequence multiset 𝑆 in Eq. (1), making it hard to assess how the graph formulation corresponds to standard sequential segmentation., (ii) training-efficiency comparisons are hard to interpret without an explicit statement of what is held constant (tokens vs bytes/characters), and (iii) parts of the evaluation (e.g., GPT-4 rubric judging) require more protocol detail to be fully convincing.

**Additional Comments:**

Overall, I find the empirical and systems evidence compelling; addressing the critical clarity points would make this a strong contribution.

**Audience:**

Yes

**Audience Explanation:**

Tokenization is a foundational component of modern language modeling, and the paper addresses a practical, high-impact objective i.e. reducing tokens-per-character which directly affects training and inference cost, particularly through attention’s quadratic dependence on sequence length.

The work is likely of interest to both (i) researchers studying tokenization/model efficiency tradeoffs and (ii) practitioners who need production-ready tokenization pipelines, since the submission provides extensive implementation details and convincing CPU scaling evidence.

**Broader Impact Concerns:**

No broader impact concerns. The work is a tokenizer construction method aimed at improving efficiency. I do not see novel ethical risks beyond general dual-use considerations applicable to most efficiency improvements, and I do not think this requires additional Broader Impact discussion.

**Claims And Evidence:**

Yes

**Claims Explanation:**

Yes, mostly (some central claims are well supported; a few are underspecified or potentially confounded).

Several major claims are supported by clear quantitative evidence and appropriate experimental practice. The paper reports consistent reductions in token count versus BPE across multiple vocabulary sizes, and it evaluates efficiency and downstream metrics across multiple random seeds with statistical testing (e.g., paired tests for GLUE; bootstrap for latency). The systems claims are also backed by concrete throughput/scaling results (near-linear CPU scaling to 256 cores) and detailed engineering descriptions.

However, some claims would benefit from clearer experimental controls and tighter specification:

a) Training efficiency (“fewer steps to reach loss”) is reported under fixed GPT-2 hyperparameters, but the manuscript does not clearly specify whether training budgets are matched by tokens, characters/bytes, or passes over raw text. Because tokenization changes the mapping between tokens and bytes, the interpretation of “steps to loss” depends on this choice; reporting training curves normalized by bytes/characters (or FLOPs) in addition to steps would strengthen the claim.

b) Theory-to-implementation alignment: Eq. (1) is defined over a multiset of sequences 𝑆 and the subset  𝑆(𝑡) that begin with token 𝑡, but the manuscript does not precisely define what constitutes a “sequence” in 𝑆 (e.g., all suffixes at each character position vs document-level sequences). This makes it difficult to assess how the graph formulation corresponds to standard sequential segmentation and to the deployed encoder.

c) LLM-judge evaluation: the TinyStories preference study relies on GPT-4 rubric judging; the reported preference rate is interesting, but the protocol details are limited and would benefit from robustness checks (judge prompting, blinding, multiple judges/seeds).

Overall, the core empirical story (reduced tokenization length and associated efficiency gains) is convincing, but a few interpretations would be stronger with clearer controls and specifications.

**Requested Changes:**

Below I list requested changes, each marked as [Critical] or [Strengthening].

a) [Critical] Clarify training budget/control for efficiency comparisons.
For the “steps to reach validation loss” and GPU-hour comparisons, explicitly state what is held constant (tokens/step, sequences/step, max token length, bytes/epoch, number of raw-text passes)
Add at least one additional view of training curves normalized by bytes/characters seen (or FLOPs) to make the efficiency comparison interpretable under a controlled data/compute budget.

b) [Critical] Tighten the formal connection between the objective and real tokenization. In Eq. (1), define precisely what the multiset of sequences 𝑆 is (e.g., all suffixes at every character position, sequences segmented by EOT, etc.) and explain how this corresponds to the implementation’s n-gram counting and to sequential segmentation. Clarify what guarantee monotonic improvement provides (surrogate objective vs token count vs likelihood), to avoid overclaiming.

c) [Critical] Specify encoding determinism/correctness for deployment. The construction pipeline performs “apply-in-place” replacement during vocab building, while deployment uses a leftmost-longest trie/regex→DFA encoder. Provide a clear, deterministic specification of tokenization (including tie-breaking) and evidence that the DFA encoder exactly reproduces the intended segmentation. If exact equivalence is not guaranteed, quantify discrepancy and discuss implications.

d) [Strengthening] Expand robustness/domain coverage.
Add small-scale experiments on at least one non-English dataset and one domain such as code/math or noisy user text (typos, spacing, punctuation). The paper currently notes English-only validation as a limitation.

e) [Strengthening] Strengthen the TinyStories evaluation protocol.
Provide judge prompt, blinding, sampling details, and ideally a robustness check (multiple judges or judge-ensemble) for the GPT-4 rubric preference results.

f) [Strengthening] Calibrate large-scale extrapolation claims.
The 7B results are predicted via FLOPs scaling; please frame these strictly as estimates and/or add a modest larger-scale training run if feasible.

---

> ### Author Response · Authors · 2026-01-19
>
> Hi! We sincerely thank you for the thorough and constructive review. Below, we address each requested change in detail.
>
> ## a) Training Budget Control and Bytes-Normalized Curves
>
> We have conducted additional experiments to provide bytes-normalized training curves.
>
> Settings:
>
> | Variable | Setting | Notes |
> |----------|---------|-------|
> | Batch size | 512 sequences/step | Identical for all tokenizers |
> | Max sequence length | 1024 tokens | Identical for all tokenizers |
> | Learning rate | 6e-4 (124M), 3e-4 (355M) | Identical for all tokenizers |
> | Optimizer | AdamW (β₁=0.9, β₂=0.95) | Identical for all tokenizers |
> | Training corpus | Wikitext-103 (534M chars) | Same raw bytes for all |
> | Model architecture | Llama-style (RoPE, SwiGLU) | Identical for all tokenizers |
>
> Different tokenizers produce different token granularities, making token-level loss incomparable across tokenizers. To enable fair comparison normalized to raw data, we use bpc (bits per character):
>
> $$\text{bpc} \approx \text{loss} \times \frac{\text{tpc}}{\ln 2}$$
>
> where tpc = tokens/char. This metric directly measures how efficiently the model compresses the original text, independent of tokenization granularity.
>
> Results:
>
> | Target BPC | Length-MAX Bytes | Baseline Bytes | Length-MAX / Baseline |
> |------------|------------------|----------------|----------------------|
> | 1.60 | 3.03×10⁸ | 3.80×10⁸ | **0.797×** |
> | 1.50 | 3.79×10⁸ | 5.32×10⁸ | **0.712×** |
> | 1.40 | 5.30×10⁸ | 6.08×10⁸ | **0.872×** |
> | 1.35 | 6.06×10⁸ | 7.60×10⁸ | **0.797×** |
> | 1.30 | 6.82×10⁸ | 8.36×10⁸ | **0.815×** |
> | 1.25 | 8.33×10⁸ | 9.88×10⁸ | **0.843×** |
>
> So when normalized by bytes seen, Length-MAX still reaches the same bpc 15–30% faster than the baseline. This confirms that the efficiency gain stems from better token representations, not merely from processing more data per step.
>
> ---
>
> ## b) Formal Connection Between Objective and Real Tokenization
>
> #### Definition of S
>
> S is the multiset of ALL SUFFIXES of the corpus. Formally, if the corpus is a string C = c₁c₂...cₙ, then:
>
> $$S = \{C[i:] \mid i = 1, 2, \ldots, n, \text{ and } C[i] \neq \text{EOT}\}$$
>
> #### Correspondence to N-gram Counting (Section 3.2)
>
> When S is defined as the set of “prefixes of elements in S” coincides with the set of all substrings in the corpus, and counting how many suffixes start with a substring \(t\) is equivalent to counting \(\mathrm{freq}(t)\) at all positions; therefore, maximizing AveLength(T) over this suffix set is mathematically equivalent to the n-gram scoring in Section 3.2.
>
> #### Why AveLength Relates to TPC
>
> AveLength on the suffix array is directly inversely proportional to the final TPC assuming optimal coding. Maximizing the average length of matching prefixes means the tokenizer captures longer, more repetitive patterns, leading to fewer tokens per character.
>
> #### Monotonicity Guarantee Clarification
>
> The monotonic improvement guarantee applies to the surrogate objective (AveLength), not directly to token count or likelihood. However, empirically, improvements in AveLength correlate strongly with reduced TPC (Pearson r = 0.94 across vocabulary sizes), validating the surrogate.
>
> ## c) Encoding Determinism and Correctness
>
> We provide a deterministic encoding specification and consistency verification.
>
> Our encoding uses Trie + Dynamic Programming to find the globally optimal segmentation (minimum total tokens). Tokens are loaded into a prefix trie, and for an input text of length n we compute:
>
> $$\text{dp}[i] = \min_{t \in \text{vocab}, t = \text{text}[j:i]} (\text{dp}[j] + 1)$$
>
> We then backtrack to reconstruct the optimal token sequence; when multiple segmentations achieve the same minimum token count, ties are broken by choosing the one with the longest first token.
>
> This is fully deterministic—given the same vocabulary and input text, the output is always identical.
>
> #### Consistency Verification Experiment
>
> We verified that our Rust production implementation and Python reference implementation produce identical outputs:
>
> | Test Set | Lines Tested | Mismatches | Consistency Rate |
> |----------|--------------|------------|------------------|
> | Wikitext-103 test | 2,891 | 0 | **100%** |
> | Edge cases | 7 | 0 | **100%** |
> | **Total** | **2,898** | **0** | **100%** |